# Effects of *Clostridium butyricum* as an Antibiotic Alternative on Growth Performance, Intestinal Morphology, Serum Biochemical Response, and Immunity of Broilers

**DOI:** 10.3390/antibiotics12030433

**Published:** 2023-02-22

**Authors:** Tiantian Yang, Mengsi Du, Jing Zhang, Baseer Ahmad, Qiang Cheng, Xiaobing Wang, Zaheer Abbas, Yucui Tong, Jinzhuan Li, Yichen Zhou, Rijun Zhang, Dayong Si

**Affiliations:** 1State Key Laboratory of Animal Nutrition, College of Animal Science and Technology, China Agricultural University, Beijing 100193, China; 2Faculty of Veterinary and Animal Sciences, Muhammad Nawaz Shareef University of Agriculture, Multan 25000, Pakistan

**Keywords:** broiler, *Clostridium butyiricum*, growth performance, intestinal morphology, immunity, serum biochemical response

## Abstract

The current study aimed to investigate the effects of *Clostridium butyiricum* on growth performance, intestinal morphology, serum biochemical response, and immunity in broiler chickens. A total of 330 commercial one-day-old, mixed-sex Ross 308 broilers were randomly divided into five treatment groups with six replicates per group. The broilers were fed the basal diet (CON), the basal diet with 150 mg/kg of aureomycin (AM), the basal diet with *C. butyricum* at 2 × 10^8^ CFU/kg (CBL), the basal diet with *C. butyricum* at 4 × 10^8^ CFU/kg (CBM), and the basal diet with *C. butyricum* at 8 × 10^8^ CFU/kg (CBH). Results showed that the final body weight (BW) (*p* < 0.01; *p* < 0.05), ADG from day 22 to 39 (*p* < 0.05), and ADG from day 1 to 39 (*p* < 0.01; *p* < 0.05) were improved in a linear and quadratic response with the inclusion of *C. butyricum*. There were no differences in feed conversion rate (FCR) among all groups (*p* > 0.05). Supplementation with *C. butyricum* quadratically reduced the crypt depth at day 21 (*p* < 0.01), linearly improved the villus height in the jejunum at day 39 (*p* < 0.001), and linearly and quadratically increased the villus height to crypt depth (V/C) ratio in the jejunum at day 21 (*p* < 0.01) and day 39 (*p* < 0.01; *p* < 0.001). Dietary *C. butyricum* affected the thymus index at day 21 and day 39 (linear, *p* < 0.01), and the bursa of Fabricius index at day 39 (quadratic, *p* < 0.05). Compared to the AM group, the serum urea contents were decreased (*p* < 0.05) but the IgG contents were increased in the CBL and CBH groups at day 21 (*p* < 0.01); in addition, serum albumin (ALB) concentrations in all the *C. butyricum*-supplemented groups (*p* < 0.01) and IgG concentrations in the CBM group were augmented at day 39 (*p* < 0.05). In conclusion, dietary *C. butyricum* could enhance growth performance by improving jejunal morphology and stimulating immunity organ development in broilers, and could be an alternative to antibiotics in poultry feeds.

## 1. Introduction

Antibiotics have been extensively applied in the livestock and poultry industry due to their beneficial effects, including growth promotion, pathogen inhibition, and disease prevention [1]. However, the wild use of antibiotics in animal production has given rise to bacteria resistance. The accumulation of antibiotic residues in feed, food, and the environment, and the transfer of resistance genes from agriculture into pathogens, is hazardous to human and animal lives [2,3,4]. Therefore, to reduce the antibiotic resistance traits in the microbial flora of farm animals, many countries ban the use of antibiotics for growth promotion [5]. Nevertheless, the increment of animal infections and the diminution of animal production are inevitable, owing to the prohibition of antibiotics as feed additives [6]. Hence, searching for growth-promoting, non-toxic, non-residual, and advantageous effect-bringing antibiotic alternatives have been of focus in the farming industry in recent years.

Direct-fed microbials, or probiotics, which are natural microorganisms, are widely used because of their safety and health benefits in livestock. They are generally accepted in the post-antibiotic era and are one of the potential representatives of antibiotic alternatives [7]. Many studies have claimed that the probiotics used for animals positively improve intestinal microbiota composition, enhance host immune response, and alleviate oxidative stress, thus favoring the healthy development of animals and further promoting their production [8,9,10]. *Clostridium butyiricum* (*C. butyiricum*) is one of the well-known probiotics that can survive and proliferate in the animal’s gut. *C. butyiricum* produces butyric acid, which is considered a source of energy for the intestinal epithelium and acts as a potent antioxidant and anti-inflammatory agent [11,12]. Meanwhile, bacteriocins, as one of the antibacterial peptides, can also be produced by *C. butyiricum* [13]. In addition, *C. butyiricum* has been reported to promote the intestinal barrier, regulate lipid metabolism, exert immune-modulatory effects, protect the host against pathogen infections and shape intestinal microflora in poultry production [14,15,16,17]. We previously isolated a *C. butyricum* strain and found that it significantly improved the growth performance, antioxidation, and immune function of broilers [18]. On the basis of our knowledge, inadequate information is known in order to express the efficacy of this *C. butyricum* strain supplementation on intestinal morphology and the serum biochemical response in broilers. We hypothesized that *C. butyricum* may stimulate the immune system, improve intestinal morphology and ultimately promote the growth performance of broilers. Consequently, the objective of this study was to assess the potential use of *C. butyricum* as an antibiotic substitute on the growth performance, intestinal morphology, immunity, and serum biochemical response of broiler chickens.

## 2. Results

### 2.1. Growth Performance

The results in Table 1 exhibit the effect of the supplementation of *C. butyricum* on the growth performance of broiler chickens. Birds fed with *C. butyricum* had a linearly and quadratically improved final BW (*p* < 0.01; *p* < 0.05). Dietary *C. butyricum* increased the average daily gain (ADG) from days 22 to 39 (linear and quadratic, *p* < 0.05) and days 1 to 39 (linear, *p* < 0.01; quadratic, *p* < 0.05).

### 2.2. Intestinal Morphology

Dietary *C. butyricum* in the feed promoted the development of the birds’ intestinal morphology (Table 2) and structure (Figure 1). During the first 21 days of the experiment, supplemental *C. butyricum* decreased the jejunal crypt depth of the birds in a quadratic response (*p* < 0.01). Compared with the AM group, the inclusion of *C. butyricum* in the feed reduced the crypt depth (*p* < 0.001). The villus height to crypt depth (V/C) ratio in the jejunum was increased with the doses of *C. butyricum* (linear, *p* = 0.01; quadratic, *p* < 0.01) on day 21. During the whole experiment, supplementation with *C. butyricum* had a positive linear effect on jejunal villus height (*p* < 0.001). Compared to the AM group, the CBH group resulted in a higher villus height (*p* < 0.001), and the CBM group decreased the crypt depth in the jejunum (*p* < 0.001). The V/C ratio in the jejunum was augmented by supplementing *C. butyricum* on day 39 (linear, *p* < 0.01; quadratic, *p* < 0.001). Compared with the AM group, the V/C ratio was higher in all the *C. butyricum*-supplemented groups (*p* < 0.001). Moreover, HE staining microscopy indicated that *C. butyricum*-treated broilers exhibited a denser and more complete jejunal villus structure than the control and antibiotic-treated broiler (Figure 1a–e).

### 2.3. Biochemical Indices in Serum

The addition of *C. butyricum* to the feed affected the serum biochemical constituents of broilers (Table 3). On day 21, compared to the AM group, the serum TG contents were lower in the CBM group, and the serum UREA contents were lower in the CBL and CBH groups (*p* < 0.05). The CREA content was increased in the AM group compared to the CON group (*p* < 0.05). On day 39, compared to the AM group, serum TC concentrations in the CBM group were higher (*p* < 0.01). Dietary *C. butyricum* in the feed increased the serum ALB contents, in comparison to the AM group (*p* < 0.01).

### 2.4. Immunity

Supplementing *C. butyricum* into the feed promoted the development of the immune organ in broilers (Table 4). Dietary *C. butyricum* increased the thymus index of broilers in a linear effect at day 21 and day 39 (*p* < 0.01). The bursa of the Fabricius index was affected by *C. butyricum* in a quadratic response at day 39 (*p* < 0.05).

As summarized in Table 5, adding dietary *C. butyricum* to the feed affected the immunoglobulin synthesis in broilers. The addition of aureomycin to the feed caused a decrease in the synthesis of IgA, IgG, and IgM. On day 21, the CBL and CBH group birds had higher serum IgG concentrations than the AM group (*p* < 0.05). On day 39, compared to the CON group, the AM group had lowered serum IgA concentrations (*p* < 0.05). The CBM group birds had a higher serum IgG level than the AM group (*p* < 0.05).

## 3. Discussion

In livestock production, probiotics used as antibiotic substitutes have been studied for decades. Probiotics exhibit positive effects on gastrointestinal morphology and the environment, increase meat quality, and produce numerous metabolites that support the animal’s immune system and growth performance; they also guard against bacterial and zoonotic infections [19,20]. *C. butyricum* is a widely studied probiotic and is of increased interest in poultry production [21,22,23]. Previously, we isolated a *C. butyricum* strain from cattle feces and found that it could improve broilers’ liver antioxidant capacity, meat quality, and fatty acid composition [24]. Our previous study showed the beneficial effects of *C. butyricum* on growth performance in broilers [18]. In the current study, *C. butyricum* has the potential to replace antibiotics in feeding production, including improving jejunal morphology, stimulating immune organ development, and maintaining serum biochemical component balance, thereby promoting the growth performance of broilers.

Supplementing feed with *C. butyricum* improved the growth performance and feed efficiency in poultry production [25,26,27]. Similarly, the addition of *C. butyricum* in the feed has been proven to prompt the growth performance of weaned piglets [28,29,30]. Consistent with previous studies, the growth performance of broilers had an upward correlation with the supplementation of *C. butyricum* in the present study. Our previous study found that broilers fed 5 × 10^8^ CFU/kg of *C. butyricum* over 1 to 42 days had the best ADG, which did not lead to a dose level-dependent result [18]. In this study, we therefore slightly modified the dosage and obtained similar results as previous research, finding that the parts of the productive parameters (final BW and ADG from 22 to 39 days and from 1 to 39 days) in the broilers that were fed with 4 × 10^8^ CFU/kg of *C. butyricum* were superior to other groups. These results suggested that higher nutrition retention was retained in the broilers due to *C. butyricum*, contributing to increased animal production and indicating that the supplemental *C. butyricum* in the feed could promote broiler growth and be used as an antibiotic substitute. The enhanced growth performance may be due to increased gastrointestinal enzyme activities, a balanced microbial population, and improved intestinal morphometric characteristics [31,32,33]. There was no significant difference in BW, ADG, ADFI, and FCR between the CON group and AM group, which indicated that the antibiotics did not show a growth-promoting effect in this study. According to previous studies, this issue remains controversial. Dietary 50 mg/kg and 150 mg/kg of aureomycin had no influence on the broilers’ growth performance (ADG and FCR) during the whole experiment, while ADFI from day 1 to 21 showed lower results in the latter [34,35]. Nevertheless, administrating broilers with 100 mg/kg of aureomycin enhanced BW and FCR [36]. These results suggested that the growth-prompting effects of antibiotics were not directly related to dose, which may be relevant considering that broad-spectrum antibiotic resistance causes increased tolerance in animals. However, in the present study, no significant differences were observed in some production performance parameters among the *C. butyricum-*supplemented groups. Similar to our study, supplementing feed with *C. butyricum* produced no significant differences in the growth performance of broilers over the whole trial period and Pekin ducks for the starter phase [37,38]. The plausible reasons are that the probiotic strains, dosage, or animals’ bioavailability affect their beneficial efficacy, suggesting a need for more studies on utilizing *C. butyricum* better. In addition, the health status of the animal itself and the administrative levels that may affect the feed conversion ratio should be considered.

As the primary site of digestion and absorption, the gut microstructure reflects the response of the intestinal tract to active substances in the feed [8]. Improved intestinal health, including “long villi” and “shallow crypts”, which are essential for enabling nutrient digestion and absorption, sustaining intestinal health and functionality, results from an integrated intestinal architecture [39]. *C. butyricum* was favorable to the intestinal morphology of various animals, including piglets, laying hens, and broilers [40,41,42]. *C. butyricum* was observed to promote the development of the villus and lower the crypt depth in mice, and the *C. butyricum*–*Lactobacillus salivarius* mixture showed better function in the villus and crypt development than *C. butyricum* alone [43]. Our study observed that *C. butyricum* increased the villus height to crypt depth ratio in a linear and quadratic dose-dependent way, and linearly augmented the villus height. It should be noted that the results found in this study suggested that a greater digestion and absorption capacity may be built in broilers, thus resulting in increased ADG and BW. These beneficial effects may be partly attributed to butyrate produced by *C. butyricum*, which facilitates intestinal epithelium growth [44]. In addition, we noticed that the crypt depth in the antibiotic group (AM) was higher than in the 2 × 10^8^ CFU/kg of *C. butyricum*-supplemented group at day 21 and in the 4 × 10^8^ CFU/kg of *C. butyricum*-supplemented group at day 39. Similar manifestations were reported in the Pekin ducks and broilers, demonstrating that supplementing aureomycin into the feed has little contribution to the villus height and crypt depth in animals [37,45]. This study and ours collectively revealed that, regardless of whether antibiotics have a beneficial effect on intestinal morphology, hosts should be administered at the proper dosage and provided with an adequate enteral environment. A large crypt implies significant tissue turnover and a considerable necessity for tissue formation. The crypt can be considered to be of as the villus factory. Furthermore, compared to other organs, the gut has a higher requirement for protein and energy. Therefore, crypt depth reduction may increase nutritional absorption [46]. Overall, our findings indicated that *C. butyricum* supplementation improves the structure of the villus height and crypt depth in the jejunum of broilers; this has positive influences on jejunal morphology.

In this study, some serum biochemical parameters were influenced by *C. butyricum*. We observed that both the serum triacylglycerol and urea contents were reduced in birds at 21 days in different dosages of *C. butyricum* compared to the AM group. Additionally, we discovered that at 39 days, broilers receiving *C. butyricum* supplementation had higher ALB concentrations than those receiving antibiotics. There will be more protein available for animal growth and development in the form of serum non-protein nitrogen (NPN), which includes serum urea, uric acid, and ammonia declines [47,48,49]. Serum ALB is responsible for carrying some substances to cells and tissues for growth and recovery, which could reflect the health status of the animals [50]. According to previous studies [51], feeding broilers 150 mg/kg of aureomycin showed no beneficial effects on serum ALB and urea. Therefore, we suggested that the *C. butyricum* was more beneficial to augment protein synthesis than antibiotics, which plays an important role in regulating nitrogen circulation as a growth promoter for broilers. Additionally, there were no significant parameters between probiotics and the control, but the values of these parameters showed no adverse effects in different treatments. There is a dispute over the regulation of probiotics on serum blood components. Some scholars indicated that probiotic supplementation did not affect the serum contents of total protein and ALB [52], which is in agreement with our results. Notwithstanding, some previous studies have shown that serum biochemical indicators are improved in probiotic-treated broilers [53,54]. The discrepancy may be due to the birds’ genetics, dosage and duration of probiotics, farming hygiene, and administration levels.

As the primary immune organs of poultry, the relative weight of the thymus, spleen, and bursa of Fabricius, reflect the immune status of chicks [55]. Probiotics have previously been proven to possess immunomodulatory activity [9,16,56]. As one of the widely used probiotics, *C. butyricum* has been reported to promote the development of the thymus, spleen, and bursa of Fabricius of broilers and laying hens [57,58,59]. In the current study, dietary *C. butyricum* linearly increased the thymus index at 21 days and 39 days and quadratically enhanced the bursa of Fabricius index in broilers at 39 days. Remarkably, the CBH group showed the best effect at the grower phase. These results suggested that *C. butyricum* stimulates the development of the broiler’s thymus at the starter phase, and a higher concentration of *C. butyricum* is better for immune organ development at the grower phase.

Serum immunoglobulins indirectly reflect the animal’s humoral immunological state, and are critical factors in regulating the immune system and resisting various ailments. More energy could be harnessed for growth and development when the host ingested enough nutrients after being well-equipped with resistance to exotic invasion. In the microenvironment of the animal gut, commensal bacteria interact with gut-associated lymphoid tissues to stimulate the production of natural antibodies in serum, such as IgA, IgG, and IgM [60]. Various studies have indicated that probiotics enhance immunoglobulin production; therefore, they served as effective feed additives for immunity improvement in livestock and poultry production [8,61,62]. *C. butyricum* stimulates immune response and activates nonspecific immunity, in order to maintain immune balance in vivo by using antimicrobial bioactive substances, such as bacteriocin [13]. Our conclusions indicated that *C. butyricum* had positive impacts on enhancing animal protection, as the IgG concentration of the birds was increased at both the starter and grower phase in *C. butyricum*-supplemented broilers compared to the antibiotic group. Previous study has demonstrated that antibiotics can alter the population structure of the indigenous microbiota, reduce bacterial diversity and disrupt intestinal homeostasis [63]. Therefore, we speculated that the in situ spontaneous interaction between the host and bacteria might be damaged due to the intervention of antibiotics and further reduce the production of immunoglobulins.

## 4. Materials and Methods

### 4.1. Bacterial Strains, Culture Conditions, and Preparation

The *C. butyricum* used in this study was isolated from cattle feces. The strain of bacteria was cultivated in Reinforced Clostridium Medium and placed in an incubator for 12 h at 37 °C in an anaerobic environment, then inoculated at 4%, placed into a 50-liter vertical fermentation tank (GuJS-50, Zhenjiang Dongfang Bioengineering Technology Co., Ltd., Zhenjiang, China) and cultured for 24 h. The culture of the strain was spray-dried directly with maltodextrin as the carrier. Lastly, the final concentrations of the prepared *C. butyricum* in the feed was 6.25 × 10^8^ CFU per gram (CFU/g) of the viable bacterium.

### 4.2. Experimental Design, Animals, and Housing

The trial was conducted in a commercial broiler-breeding corporation (Qilibao Chicken Farm, Hebi, Henan, China). All procedures performed were approved by the Ministry of Science and Technology (Beijing, China). The birds were fed in three-dimensional three-tier cages. Birds were exposed to 24 h of light during the first 7 days, then 22 h of light, and 2 h of dark thereafter during the experiment. The ambient temperature was maintained at 33 °C for a week, decreased by 0.5 °C every day to 29.5 °C and then reduced by 0.3 °C each day to a final temperature of 22 °C. The relative humidity in the bird’s house was maintained at 60–70% within the first 2 weeks and then at 50% afterward. All chickens were vaccinated against infectious bronchitis (1 and 10 days of age) and Newcastle disease (10 days of age) (Nobilis^®^ ND LaSota, Intervet International, Boxmeer, The Netherlands). The feed and drinking water were available ad libitum during the entire period.

A total of 330 commercial one-day-old, mixed-sex Ross 308 broilers were randomly divided into five treatment groups. Each treatment consisted of 6 replicates with 11 birds per replicate. The broilers were fed with the basal diet (CON), the basal diet with 150 mg/kg of aureomycin (AM), the basal diet with *C. butyricum* at 2 × 10^8^ CFU/kg (CBL), the basal diet with *C. butyricum* at 4 × 10^8^ CFU/kg (CBM), and the basal diet with *C. butyricum* at 8 × 10^8^ CFU/kg (CBH). The experiment lasted for 39 days. All birds were provided with a 2-phase feeding program (Days 0–21 as the starter phase, Days 22–39 as the finisher phase). The basal diet was formulated to meet National Research Council (NRC, 1994) requirements. The feed composition and chemical analysis of the basal diet is shown in Table 6.

### 4.3. Growth Performance

The initial live weights of the birds were measured at the start of the study. After a 12-hour feed withdrawal, the broilers’ feed intake and body weight (BW) in each pen were measured on days 21 and 39. The average daily gain (ADG), average daily feed intake (ADFI), and feed conversion ratio (FCR) were calculated for starters, growers, and overall periods. Mortality and health status were observed and recorded daily throughout the experiment.

### 4.4. Sample Collection

Blood samples were collected on the days 21 and 39 days (1 bird per replicate). Birds were selected with weights close to the average weight from each treatment in two periods (days 21 and 39). Blood samples were collected from the wing vein in a disposable vacuum blood collection tube individually and centrifuged at 845× *g* for 15 min at 4 °C, which was stored at −20 °C for assaying. After bleeding, birds were euthanized by cervical dislocation, exsanguinated, scalded, and then de-feathered. Additionally, for intestinal histological examination, approximately 2 cm of jejunum was collected, washed with sterile saline solution, and preserved in 4% paraformaldehyde. Furthermore, lymphoid organ weights (spleen, bursa of Fabricius, and thymus) on days 21 and 39 were weighed individually and represented as a proportion of BW.

### 4.5. Intestinal Morphology Analysis

The method of intestinal morphology analysis was slightly modified according to the previous study [8]. Approximately 2-cm segments of jejunum were stored in 4% paraformaldehyde. The tissues were then cut into 5 μm sections, routinely dehydrated in ethanol, dipped in wax, embedded in paraffin, dewaxed with xylene, hydrated, and stained with hematoxylin-eosin (HE). The villus height and crypt depth were observed by a live-cell station (Leica DMI8, Weztlar, German). Ten fields of vision were randomly picked from each segment to assess the villus height and crypt depth.

### 4.6. Immunological and Biochemical Analysis

Specific ELISA kits (Leadman Group Co., Ltd., Beijing, China) were used to assess the contents of serum immunoglobulins (IgA, IgM, and IgG), following the manufacturer’s instructions. Biochemical indices such as glucose (GLU), total cholesterol (TC) and triacylglycerol (TG), total protein (TP), albumin (ALB), urea (UREA), and creatinine (CREA) in the serum were measured by an automatic biochemical analyzer (RA-1000, Bayer Corp., Tarrytown, NY, USA).

### 4.7. Statistical Analysis

Data were subjected to one-way ANOVA (Tukey’s-test) using IBM SPSS Statistics 20 statistical package (SPSS Inc., Chicago, IL, USA) as a completely randomized design with a pen (cage) as the unit. The results were expressed as means. Differences were considered significant at *p* < 0.05. Orthogonal contrasts were used to determine the linear and quadratic effects of the increasing levels of supplementation of *C. butyricum*.

## 5. Conclusions

Our study concluded that *C. butyricum*, as a natural feed additive in the broiler diet, promoted the average daily gain and final body weight, improved jejunal morphology and prompted immune organ growth without affecting most of the serum biochemical parameters. Moreover, *C. butyricum* could maintain the host’s innate immune system integrity. In addition, the aureomycin showed no significant improvement effect on the broilers, and the *C. butyricum*-supplemented groups performed better in part of serum biochemical and immune indices than the aureomycin group in the present study. Consequently, these findings demonstrated that *C. butyricum* could be used as an alternative to in-feed antibiotics as a growth promoter in broiler production. Further research is necessary to explore the growth-prompting mechanism at the level of gene regulation.

## Figures and Tables

**Figure 1 antibiotics-12-00433-f001:**
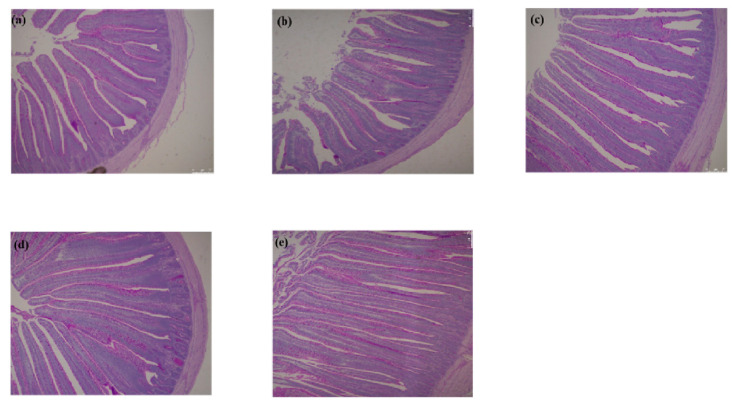
Effects of *Clostridium butyricum* on the jejunal histomorphology in broilers. (**a**–**e**) jejunum biopsies of CON, AM, CBL, CBM, and CBH group birds on day 39. The broilers were fed with CON, the basal diet; AM, the basal diet with 150 mg/kg of aureomycin; CBL, the basal diet with *C. butyricum* at 2 × 10^8^ CFU/kg; CBM, the basal diet with *C. butyricum* at 4 × 10^8^ CFU/kg; and CBH, the basal diet with *C. butyricum* at 8 × 10^8^ CFU/kg.

**Table 1 antibiotics-12-00433-t001:** Effects of *C. butyricum* on growth performance ^1^.

Item	Treatment ^2^					SEM ^3^	*p*-Value	Response to CB	
	CON	AM	CBL	CBM	CBH			Linear	Quadratic
Initial BW (1 day, kg)	47.67	47.35	47.53	47.27	47.77	0.089	0.369	0.973	0.161
Final BW (39 day, kg)	1620.17 ^b^	1662.37 ^ab^	1667.30 ^ab^	1733.02 ^a^	1693.98 ^ab^	12.157	0.038	0.005	0.047
1 to 21 days									
ADG (g/day)	23.79	23.53	24.01	23.90	24.33	0.226	0.873	0.545	0.848
ADFI (g/day)	39.15	36.43	39.68	39.45	40.73	0.492	0.063	0.370	0.736
FCR (g/g)	1.65	1.55	1.65	1.65	1.68	0.018	0.218	0.648	0.755
22 to 39 days									
ADG (g/day)	59.61 ^b^	62.27 ^ab^	61.97 ^ab^	65.78 ^a^	63.08 ^ab^	0.660	0.045	0.014	0.044
ADFI (g/day)	107.34	111.78	110.97	111.98	111.20	1.559	0.897	0.402	0.510
FCR (g/g)	1.80	1.79	1.79	1.70	1.76	0.017	0.402	0.231	0.351
1 to 39 days									
ADG (g/day)	40.32 ^b^	41.41 ^ab^	41.53 ^ab^	43.23 ^a^	42.21 ^ab^	0.312	0.037	0.005	0.045
ADFI (g/day)	74.68	73.27	75.66	78.06	77.65	1.125	0.663	0.359	0.799
FCR (g/g)	1.84	1.77	1.82	1.80	1.84	0.022	0.819	0.868	0.583

^a,b^ Means within a row with different letters differ (*p* < 0.05). ^1^ BW, body weight. ADG, average daily gain. ADFI, average daily feed intake. FCR, feed conversion rate. Results are given as means, *n* = 6. ^2^ The broilers were fed with CON, the basal diet; AM, the basal diet with 150 mg/kg of aureomycin; CBL, the basal diet with *C. butyricum* at 2 × 10^8^ CFU/kg; CBM, the basal diet with *C. butyricum* at 4 × 10^8^ CFU/kg; and CBH, the basal diet with *C. butyricum* at 8 × 10^8^ CFU/kg. ^3^ SEM, standard error of the mean.

**Table 2 antibiotics-12-00433-t002:** Effects of *C. butyricum* on jejunum morphology of broilers ^1^.

Item	Treatment ^2^					SEM ^3^	*p* Value	Response to CB	
	CON	AM	CBL	CBM	CBH			Linear	Quadratic
21 days									
Villus height (μm)	1067.87	1201.50	1215.06	1361.43	1339.02	44.471	0.225	0.050	0.420
Crypt depth (μm)	188.15 ^ab^	210.80 ^a^	117.94 ^c^	151.12 ^bc^	151.61 ^bc^	7.406	<0.001	0.100	0.002
V/C (μm/μm)	5.76 ^b^	5.80 ^b^	10.30 ^a^	8.98 ^a^	9.01 ^a^	0.430	<0.001	0.010	0.003
39 days									
Villus height (μm)	1070.10 ^b^	1182.72 ^b^	1367.54 ^b^	1335.96 ^b^	1689.48 ^a^	50.000	<0.001	<0.001	0.739
Crypt depth (μm)	165.52 ^abc^	183.78 ^ab^	146.12 ^bc^	138.52 ^c^	204.65 ^a^	6.258	<0.001	0.025	<0.001
V/C (μm/μm)	6.50 ^b^	6.55 ^b^	9.37 ^a^	9.77 ^a^	8.27 ^a^	0.304	<0.001	0.004	<0.001

^a,b,c^ Means within a row with different letters differ (*p* < 0.05). Results are given as means, *n* = 6. ^1^ V/C, the villus height to crypt depth (V/C) ratio. ^2^ The broilers were fed with CON, the basal diet; AM, the basal diet with 150 mg/kg of aureomycin; CBL, the basal diet with *C. butyricum* at 2 × 10^8^ CFU/kg; CBM, the basal diet with *C. butyricum* at 4 × 10^8^ CFU/kg; and CBH, the basal diet with *C. butyricum* at 8 × 10^8^ CFU/kg. ^3^ SEM, standard error of the mean.

**Table 3 antibiotics-12-00433-t003:** Effects of *C. butyricum* on serum biochemical indices of broilers ^1^.

Item	Treatment ^2^					SEM ^3^	*p* Value	Response to CB	
	CON	AM	CBL	CBM	CBH			Linear	Quadratic
21 days									
GLU (mmol/L)	10.67	10.98	10.50	10.50	9.45	0.175	0.055	0.041	0.255
TC (mmol/L)	3.34	4.12	3.53	3.24	3.63	0.120	0.156	0.658	0.725
TG (mmol/L)	0.32 ^ab^	0.39 ^a^	0.30 ^b^	0.35 ^ab^	0.35 ^ab^	0.010	0.048	0.174	0.737
TP (g/L)	26.50	23.12	24.80	26.08	24.63	0.565	0.361	0.470	0.925
ALB (g/L)	12.00	10.65	12.28	12.15	11.33	0.257	0.229	0.418	0.351
UREA (mmol/L)	0.53 ^ab^	0.59 ^a^	0.41 ^b^	0.49 ^ab^	0.43 ^b^	0.020	0.011	0.303	0.440
CREA (μmol/L)	9.05 ^b^	11.92 ^a^	11.15 ^ab^	11.03 ^ab^	10.45 ^ab^	0.297	0.022	0.142	0.036
39 days									
GLU (mmol/L)	10.50	11.09	10.83	10.86	9.59	0.182	0.066	0.107	0.036
TC (mmol/L)	3.57 ^ab^	2.95 ^b^	3.32 ^ab^	4.24 ^a^	3.43 ^ab^	0.122	0.008	0.607	0.223
TG (mmol/L)	0.37	0.45	0.42	0.38	0.39	0.014	0.401	0.814	0.471
TP (g/L)	31.85	27.15	35.73	32.05	32.87	0.966	0.071	0.948	0.486
ALB (g/L)	11.38 ^a^	9.48 ^b^	12.38 ^a^	11.73 ^a^	11.83 ^a^	0.252	0.001	0.721	0.311
UREA (mmol/L)	0.44	0.59	0.55	0.42	0.47	0.022	0.069	0.798	0.523
CREA (μmol/L)	9.75	11.43	12.17	10.57	10.35	0.288	0.055	0.935	0.025

^a,b^ Means within a row with different letters differ (*p* < 0.05). Results are given as means, *n* = 6. ^1^ GLU, glucose; TC, total cholesterol; TG, total triacylglycerol; TP, total protein; ALB, albumin; UREA, urea; CREA, creatinine. ^2^ The broilers were fed with CON, the basal diet; AM, the basal diet with 150 mg/kg of aureomycin; CBL, the basal diet with *C. butyricum* at 2 × 10^8^ CFU/kg; CBM, the basal diet with *C. butyricum* at 4 × 10^8^ CFU/kg; and CBH, the basal diet with *C. butyricum* at 8 × 10^8^ CFU/kg. ^3^ SEM, standard error of the mean.

**Table 4 antibiotics-12-00433-t004:** Effects of *C. butyricum* on immune organ index of broiler chickens.

Item	Treatment ^1^					SEM ^2^	*p*-Value	Response to CB	
	CON	AM	CBL	CBM	CBH			Linear	Quadratic
21 days									
Thymus (g/kg)	2.41 ^b^	2.75 ^ab^	2.37 ^b^	3.22 ^a^	3.23 ^a^	0.112	0.012	0.003	0.928
Spleen (g/kg)	1.00	1.26	1.06	1.07	1.08	0.034	0.139	0.441	0.761
Bursa of Fabricius (g/kg)	2.50	2.67	2.39	2.33	2.57	0.067	0.532	0.882	0.277
39 days									
Thymus (g/kg)	2.17 ^b^	2.55 ^ab^	2.17 ^b^	2.61 ^ab^	3.10 ^a^	0.099	0.008	0.001	0.197
Spleen (g/kg)	2.19	1.96	1.90	1.82	2.34	0.073	0.124	0.631	0.023
Bursa of Fabricius (g/kg)	1.11 ^ab^	1.31 ^ab^	1.02 ^b^	1.05 ^b^	1.40 ^a^	0.051	0.048	0.072	0.049

^a,b^ Means within a row with different letters differ (*p* < 0.05). Results are given as means, *n* = 6. ^1^ The broilers were fed with CON, the basal diet; AM, the basal diet with 150 mg/kg of aureomycin; CBL, the basal diet with *C. butyricum* at 2 × 10^8^ CFU/kg; CBM, the basal diet with *C. butyricum* at 4 × 10^8^ CFU/kg; and CBH, the basal diet with *C. butyricum* at 8 × 10^8^ CFU/kg. ^2^ SEM, standard error of the mean.

**Table 5 antibiotics-12-00433-t005:** Effects of *C. butyricum* on serum immune parameters (g/L) of broiler chickens.

Item	Treatment ^1^					SEM ^2^	*p* Value	Response to CB	
	CON	AM	CBL	CBM	CBH			Linear	Quadratic
21 days									
IgA	1.02	0.89	0.94	0.99	0.99	0.026	0.619	0.882	0.610
IgG	6.61 ^ab^	5.91 ^b^	7.27 ^a^	6.91 ^ab^	7.65 ^a^	0.172	0.009	0.089	0.911
IgM	0.90	0.76	0.90	0.99	0.95	0.027	0.069	0.360	0.803
39 days									
IgA	1.25 ^a^	0.86 ^b^	1.05 ^ab^	1.09 ^ab^	0.96 ^ab^	0.040	0.019	0. 035	0.621
IgG	7.65 ^ab^	6.42 ^b^	7.63 ^ab^	8.34 ^a^	7.54 ^ab^	0.186	0.015	0.808	0.303
IgM	0.95	0.85	0.98	0.86	0.82	0.022	0.091	0.031	0.462

^a,b^ Means within a row with different letters differ (*p* < 0.05). Results are given as means, *n* = 6. ^1^ The broilers were fed with CON, the basal diet; AM, the basal diet with 150 mg/kg of aureomycin; CBL, the basal diet with *C. butyricum* at 2 × 10^8^ CFU/kg; CBM, the basal diet with *C. butyricum* at 4 × 10^8^ CFU/kg; and CBH, the basal diet with *C. butyricum* at 8 × 10^8^ CFU/kg. ^2^ SEM, standard error of the mean.

**Table 6 antibiotics-12-00433-t006:** Compositions and chemical analysis of basal diets for broilers (%).

Item (%)	Starter (Day 1–21)	Finisher (Day 22–39)
Ingredients		
Corn	54.70	56.70
High protein soybean meal	34.70	25.90
CaHPO_4_	1.50	1.20
Limestone	1.20	1.20
Soybean oil	0.90	3.50
Wheat	5.00	7.50
Chicken bone meal	0.00	2.00
Premix ^1^	2.00	2.00
Total	100.00	100.00
Chemical composition analyzed ^2^		
ME, Mcal/kg	2.95	3.10
Dry matter	87.35	87.70
Crude protein	22.00	20.50
Calcium	0.90	0.87
Total phosphorus	0.61	0.57
Non-phytate phosphorus	0.45	0.45
Lysine	1.37	1.20
Methionine	0.67	0.60
Methionine + Cysteine	0.96	0.82
Threonine	0.95	0.84

^1^ Premix provided the following nutrients per kilogram of diet: Fe, 111 mg; Cu, 10 mg; Mn, 128 mg; Zn, 142 mg; NaCl, 3 g; L-lysine HCl, 2.4 g; DL-methionine 1.4 g; vitamin A, 14,000 IU; vitamin D3, 6000 IU; vitamin E, 70 mg; vitamin K3, 4 mg; vitamin B1, 7 mg; vitamin B2, 13 mg; vitamin B6, 13 mg; vitamin B12, 29 μg; choline, 1835 mg; folic acid, 3 mg; nicotinic acid, 93 mg; and pantothenic acid, 27 mg. ^2^ The chemical composition of feed in this experiment was calculated.

## Data Availability

The datasets during and/or analyzed during the current study are available from the corresponding authors upon reasonable request.

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
