# Peer review of "Effects of Clostridium butyricum as an Antibiotic Alternative on Growth Performance, Intestinal Morphology, Serum Biochemical Response, and Immunity of Broilers"

_antibiotics, 2023, doi:10.3390/antibiotics12030433_

Round 1
Reviewer 1 Report
Thanks for your manuscript. There are critical issues inside methods section.
1.please revise the english gramer.
2. In my opinion, it is difficult to get clear and sound conclusion for 2 reasons:1) no challenge of any bacteria , thus, it is difficult to answer the aim of this study( find alternate to Antibiotics).2) the authors used mixed sex, thus it is very hard to get sound data for gut and blood indices. Next time please collect samples from one sex.
Last, in my opinion, the parameters adopted in the present study is not sufficient to answer the aim of this manuscript.
Author Response
Dear reviewer,
Thank you for your kind and careful reviewing the manuscript (Antibiotics-2151899), titled “Effects of Clostridium butyricum as an antibiotic alternative on growth performance, intestinal morphology, serum biochemical response, and immunity of broilers”.
Based on these comments and suggestions, we have revised all the problems you proposed and improved the manuscript seriously, incuding all the inappropriate contents.
All the responces to the questions were listed below.
Point 1: please revise the English grammar.
Response 1: We have checked the English grammar and corrected some errors.
Point 2: no challenge of any bacteria, thus, it is difficult to answer the aim of this study( find alternate to Antibiotics).
Response 2: Although we have not imposed any pathogens challenge on broilers in this study, positive results (part of growth performance parameters, intestinal morphology and immune organ index) were shown in probiotic-supplemented groups, which were superior to the control group and antibiotic group. In agreement with some previous studies, C. butyricum has the potential to be an antibiotic alternative due to improved growth performance, intestinal morphology and immune function.
Point 3: the authors used mixed sex, thus it is very hard to get sound data for gut and blood indices.
Response 3: Firstly, we used mixed sex broilers as the objective of this experiment on the basis of some reports and our previous studies, which was in line with their experimental designs. Secondly, some studies have shown that there are no significant differences in serum biochemical indexes of poultry between broilers of different sex. The following are some relevant articles:
- https://www.researchgate.net/publication/289282578_Effects_of_dietary_CrCl3_supplementation_on_some_serum_biochemical_markers_in_broilers_Influence_of_season_age_and_sex.
- https://www.sid.ir/paper/586931/en.
- https://europepmc.org/article/med/24163966.
Thank for your careful review and relevant comments again.
Best wishes and regards,
Reviewer 2 Report
That’s a very nice and well-written paper, with a clearly stated problem and objective, correctly prepared results, good discussion and conclusion.
The topic is interesting and thus the paper is worth of publication. I have a few questions (in the attached word doc. antibiotics-2151899-.docx) for clarification and some suggestions.
Comments are also below
Abstract
L17: Please mention the number of replications.
L20-23: Please mention something about feed per gain (FCR)
Introduction
L41: Containing? growth promotion. You mean including?
L45-47: rewrite this section
L66: C. butyricum NF. What does NF mean?
Results
L120: led a higher? Had a higher?
Discussion
L145: more nutrition retention was retained? You mean ‘higher nutrient retention?’
L149: Please check these two references.
reference 31: they used a plant that has bioactive components not a bacteria.
reference 33: probiotic did not affect growth performance, incidence of necrotic enteritis or caecal microbiota
L168: dose-dependent
L170: constructed in broilers? Please use an appropriate verb other than constructed
L183: downregulated? Were reduced you mean?
L201: the (related weight) correct to “the relative weight”
Materials and method
L240: how was the trial conducted in a commercial hatchery?
L243: of light
Author Response
Dear reviewer,
Thank you for your kind and careful reviewing the manuscript (Antibiotics-2151899), titled “Effects of Clostridium butyricum as an antibiotic alternative on growth performance, intestinal morphology, serum biochemical response, and immunity of broilers”.
Based on these comments and suggestions, we have revised all the problems you proposed and improved the manuscript seriously, incuding all the inappropriate contents.

Reviewer 3 Report
Dear Editor and Authors;
I have reviewed the manuscript entitled " Effects of Clostridium butyricum as an antibiotic alternative on growth performance, intestinal morphology, serum biochemical response, and immunity of broilers" (antibiotics-2151899). Research on the subject of the study fits within the journal's scope. An adequate amount of information has been provided in the Introduction, and the researchers have clearly stated the objectives of their study. In the results and discussion sections, sufficient information is provided to inform the reader about the authors' materials and methods used in their study. There are, however, a few minor points to be addressed before publication.
Best regards.
Minor comments
According to journal writing rules, the abstract has a relatively high number of words. I believe that the authors will write an abstract that is short but effective in the number of words that it contains.
It is recommended that the tables at the end of the text be placed in the “Results“ section.
L14: "(C. butyiricum)" should be omitted.. In binomial nomenclature, the genus can be abbreviated immediately after it is first used.
L36-37: It is recommended that keywords be sorted alphabetically.
L100-101: Footnotes provide a clear description of the content of the trial groups, as in the Abstract (L17-20).
L115: Please use “Table 6” instead of “table 6”.
L155: Please use “starter phase” instead of “start phase”.
L250: Please italicize “ad libitum”.
L295: Please use “indices” instead of “Indices”.
L484: Please use “indices” instead of “index”.
L490, L495: Please use “C. butyricum” instead of “Clostridium butyricum”.
Author Response

(The authors gave the same response as above.)

Round 2
Reviewer 1 Report
Thanks for your revised manuscript. However, Still many vital mistakes in this manuscript beside no challenge and use unsexed birds. For example, if you check FCR you can find the lowest one with 1.77. Nowadays FCR values rang between 1.3~1.4 under normal conditions ( without stress / challenges).
Author Response

(The authors gave the same response as above.)
